# Non-Thermal Atmospheric Plasma for Microbial Decontamination and Removal of Hazardous Chemicals: An Overview in the Circular Economy Context with Data for Test Applications of Microwave Plasma Torch

Yovana Todorova [1,2,*], Evgenia Benova [2], Plamena Marinova [2,3], Ivaylo Yotinov [1,2], Todor Bogdanov [2,4] and Yana Topalova [1,2]

1 Faculty of Biology, Sofia University "St. Kliment Ohridski", 8 Dragan Tsankov Blvd., 1164 Sofia, Bulgaria; ivaylo_yotinov@uni-sofia.bg (I.Y.); ytopalova@uni-sofia.bg (Y.T.)
2 Clean & Circle Center of Competence, Sofia University, 1164 Sofia, Bulgaria; ebenova@uni-sofia.bg (E.B.); plamena_dragozova@ltu.bg (P.M.); tbogdanov@medfac.mu-sofia.bg (T.B.)
3 Faculty of Forest Industry, University of Forestry, 1756 Sofia, Bulgaria
4 Department of Medical Physics and Biophysics, Faculty of Medicine, Medical University of Sofia, 1431 Sofia, Bulgaria
* Correspondence: yovanatodorova@biofac.uni-sofia.bg

**Abstract:** The transformation of our linear "take-make-waste" system to a cyclic flow of materials and energy is a priority task for society, but the circular use of waste streams from one industry/sector as a material input for another must be completely safe. The need for new advanced technologies and methods ensuring both microbiological safety and the removal of potential chemical residues in used materials and products is urgent. Non-thermal atmospheric plasma (cold atmospheric plasma—CAP) has recently attracted great research interest as an alternative for operative solutions of problems related to safety and quality control. CAP is a powerful tool for the inactivation of different hazardous microorganisms and viruses, and the effective decontamination of surfaces and liquids has been demonstrated. Additionally, the plasma's active components are strong oxidizers and their synergetic effect can lead to the degradation of toxic chemical compounds such as phenols and azo-dyes.

**Keywords:** non-thermal atmospheric plasma; circle economy; safety; microbial decontamination; removal of hazardous chemicals

## 1. Introduction

Our contemporary society is facing many environmental challenges due to the unprecedented level of industrialization, urbanization, and exponential growth of the human population. The increasing generation and disposal of wastes, environmental pollution, source depletion, biodiversity loss, climate changes, and growing energy demand are global threats for our sustainable development, and require a fundamentally new rethinking to reduce the consumer footprint on nature [1–4]. The transformation of our linear "take-make-waste" system to the cyclic flow of materials and energy is urgent, and this is a key driver for rapid development and promotion of the circular economy concept. The circular economy has the potential to overcome challenges by following three general principles: "(1) preserve and enhance natural capital by controlling finite stocks and balancing renewable resource flows; (2) optimize resource yields by circulating products, components and materials in use at the highest utility; (3) foster system effectiveness by revealing and designing out negative externalities" [5,6]. The most widely applicable aspects of the circular economy include eco-design for sustainability, extension of product life cycle, and implementation of "reduce, reuse and recycle" option in waste management [7,8]. However, the closing of loops in different industrial sectors by circulation, reuse, and recycling of

materials and products bring to the fore other key issues—those related to safety and the prevention of risks from secondary contamination. Ensuring a high level of uncompromising safety in circular economy practices is a priority task and an important keystone for the achievement of high social acceptance and positive user perceptions and attitudes.

When discussing safety issues, it must be taken into account that safety includes both microbiological and chemical aspects. The context of the circular economy is no exception-we must be sure that the use of waste streams from one industry/sector as a material input for another is completely safe and the possibility for crossover of chemical or microbiological hazards is eliminated/minimized. The problem is escalating nowadays with the enormous production and use of chemicals, resulting in a global scale of chemical pollution, and the pandemic distribution of unknown viruses and multidrug-resistant bacteria [9,10]. It is clear that we urgently need new advanced approaches ensuring both microbiological safety and the removal of potential chemical residues in different materials and products, with quick achievement of the high safety level, easy operation, lack of residual toxicity, and wide use with no application restrictions [11–13]. Facing these scientific and practical challenges, it seems that one well-known physical phenomenon—plasma—and some recently developed plasma-based technologies can effectively respond to these requirements.

Plasma is a quasi-neutral system in a gaseous or fluid-like form that can be artificially generated in an electromagnetic field and a flow of neutral gases such as helium, argon, nitrogen, oxygen, or atmospheric air. It contains a mixture of radicals, $H_2O_2$, $O_3$, ultraviolet radiation, charged particles, exited metastable atoms, and electric fields [14,15]. The plasma active components individually are well-known sterilization agents, and as expected, their combination in plasma has a strong synergistic effect and provides high bactericidal efficiency with low costs, timesaving, and non-toxicity.

Based on the temperature of particles, there are two types of plasmas: high-temperature plasmas (fusion plasmas) and low-temperature plasmas (non-thermal or cold plasmas) [14]. Each type of plasma has a specific application and benefits but in the field of biosafety and microbiological control of different materials and products, the leading role of non-thermal plasma is undoubted, especially the type generated at atmospheric pressure [11,15–27]. Its main advantage is a simple and less expensive plasma source operating in an open space without thermal damage on treated materials. Non-thermal plasmas may be generated by direct current (DC) discharge, radio frequency discharge, dielectric barrier discharge, pulsed power, and surface wave (microwave) discharge. Microwave plasma not in equilibrium, the electron energy distribution function is non-Maxwellian, and the temperature of the heavy particles is much lower than the electron temperature [28]. Non-equilibrium, non-thermal atmospheric pressure plasma (also called cold atmospheric plasma—CAP) is suitable to treat living tissues, heat-sensitive materials, foods, and bio-compatible materials, and has wide use in medicine, the agro-food industry, and microbiological control. The use of CAP is not limited to an alternative sterilization or therapeutic technique for biomedical application—the plasma-based approach has an important role in the solution of some critical environmental issues. Recent data show its significant potential as an effective tool in pollution control and the treatment of different polluted sources. Looking at the wide field of plasma applications from a different perspective, it is clear that this huge potential can successfully contribute to improving the microbiological and chemical safety of products and materials in the circular economy.

This paper presents a summary of the different applications of CAP with some specific case studies from our work with cold argon plasma in the context of safety in the circular economy. The paper is structured as follows. In Section 2, we represent a mini-review and resume of our experiments and results for bacterial decontamination of surfaces and liquids with non-thermal non-equilibrium plasma; a general characterization of the used plasma source, its active components, and the experimental set-up are also presented. Section 3 discusses some aspects of plasma removal of trivial and recalcitrant organic compounds. The last section is a brief resume and discussion of the potential of cold argon atmospheric plasma for future uses in a circular economy context.

## 2. Low-Temperature Non-Equilibrium Plasma for Microbial Decontamination

### 2.1. A Brief State of the Art

For convenience, the specific fields of CAP where its germicidal effect is used can be generalized as follows: (1) plasma-aided medical therapies, (2) plasma-assisted dentistry, and (3) plasma-based decontamination and sterilization for microbiological control in agriculture, the food industry, microbiology, water treatment, etc.

The application of plasma technologies as an alternative advanced therapeutic modality in human health care is well presented in some recent reviews and studies and we will not discuss it in detail [29–35]. In this section, we will focus on a CAP use for bio-decontamination and sterilization of different surfaces and liquids because this application has the potential to respond to circular economy needs. This field covers various aspects of well-known microbiological processes of the elimination, removal, killing, or deactivation of live forms or biological agents.

The interaction of CAP with living objects, bacteria, spores, and viruses has been extensively studied in the last decades, and basic principles have been defined [36–39]. The main attention has been paid to plasma's killing effect on bacteria (both Gram-positive and Gram-negative), bacterial spores, biofilms, and fungi. A significant efficiency of treatment has been gained with different CAP generated configurations against a large number of pathogens with high-risk profiles: *Escherichia coli*, *Listeria monocytogenes*, *Salmonella enterica*, *Staphylococcus* sp., *Pseudomonas aeruginosa*, *Bacillus* sp.; and spores such as *Candida albicans* [40–47]. The bactericidal effect of CAP is clearly expressed in the treatment of biofilms on different surfaces and bacterial suspensions in liquids. Viruses and prions have been studied to a lesser extent, but in the last two years, this gap has begun to fill intensively due to the COVID-19 crisis [48–50]. Many studies have demonstrated the ability of non-thermal plasma to be an effective virucidal agent. The plasma treatment reduces virus burdens on contaminated surfaces and airborne viruses. and can be applied for successful prevention and interference of virus replication [51].

The exact mechanism of plasma interaction with biological objects has not been fully understood. The main reason is the enormous diversity of all major participants in this complex process:

- Very diverse biological targets—for example, the interaction of plasma with prokaryotic and eukaryotic cells is probably based on different mechanisms [52]; Gram-negative and Gram-positive bacteria also have a different sensitivity to plasma treatment [53]. Cell organization in biofilms assumes a new high hierarchical level of interaction with external impacts. Bacterial metabolic status and cell morphology are important factors for inactivation efficiency [54];
- Many active plasma components with different effects on biological objects—UV affects the dimerization of thymine bases in DNA; charges particles rupture the cell membranes; reactive species of oxygen and nitrogen have a strong oxidative effect also on bacterial membranes [55], and on viral ability to bind with host receptors and to enter into host cells [51];
- Physical and chemical properties of plasma depend on many factors: the type of configuration, mode of operation, environment (air, water, surfaces), and gases used can play a significant role in the presence, proportion, and concentration of plasma inactivation agents and reactive species. It seems that the application and efficiency of plasma depend on these specific characteristics of the devices, making it challenging to study the exact mode of action and mechanism of interaction [53].

Some of the hypotheses for the inactivation mechanism of CAP on bacteria are summarized by Fernandez and Thompson as follows [56]: the destroying effect of UV irradiation on DNA according to its power density and wavelength range [55]; damage from the diffusion of reactive species through membranes and their reaction with cell macromolecules [57]; the etching effect on the cell surface by direct bombardment with free radicals [58]; erosion of the microorganisms through intrinsic photodesorption by UV [59]. The last two lead

to a massive leakage of cell content, a strongly negative effect on cell adhesion at biofilm growth, and the interruption or inhibition of biofilm quorum sensing systems [54].

In the field of agriculture and food processing, the search for new promising and effective "cleaning" tools with a multipurpose application based on cold atmospheric plasma has also been the subject of many types of research [16,24,60–64]. The growing need for "green", un-, or less processed food with high quality, safety, and preserved taste and nutrients leads to a very attractive rating of CAP as a powerful non-thermal sterilization technique. The recent studies are aimed at the treatment of food products, food processing devices, packaging materials, functionality modification of food materials, and removal of agrochemical residues [22,65–67]. Non-thermal atmospheric plasma is a promising decontamination technology for the inactivation of bacteria, yeasts, molds, and fungal and bacterial spores both on the foods and on the abiotic surfaces of packages or processing equipment [68]. The obtained results reveal promising opportunities to use CAP to ensure safety, especially in the case of the entry of waste streams in different stages of processing. The reuse of composted food wastes or activated sludge as fertilizer in agriculture, for example, may increase microbiological risk, while appropriate plasma treatment of final product may eliminate the hazard. Additionally, plasma can be used for alternative food processing to obtain the desired color of meat products without adding chemicals, to improve the extraction of essential oils, or to extend the life of some products [27,68].

In the next subsections, our work in this field is presented as an example of the successful use of cold argon plasma for bacterial inactivation. We studied the sterilization effect of CAP on bacteria (both Gram-negative and Gram-positive) grown in biofilms on surfaces and in suspensions. One of the supposed mechanisms of plasma interaction with bacterial cells—serious damage to cell surfaces incompatible with life—was also shown.

### 2.2. Case Study—Bactericidal Effect of Cold Argon Plasma on Surfaces and in Liquids

#### 2.2.1. Characterization of the Plasma Generation System

Non-equilibrium, non-thermal atmospheric pressure plasma can be generated by different sources [69,70]. We use a surface wave discharge that produced the plasma by an electromagnetic wave travelling along with the plasma-dielectric interface. The plasma is produced in argon at atmospheric pressure by a surfatron type electromagnetic wave launcher operating at 2.45 GHz. Discharge conditions are: plasma radius 0.05 cm ÷ 0.1 cm; gas flow rate 3.2 L/min. Treatment times during all the investigations are significantly short—less than a minute—and the input wave powers are under 20 W. The low gas temperature allows plasma to be in close contact with the treated samples.

#### 2.2.2. Experimental Design

The design for the study of the sterilization effect of the cold argon plasma torch and specifics of the used microorganisms, media, and experimental conditions are presented in Table 1. More detailed information for the used bacterial strains is given in Marinova et al. and Todorova et al. [15,71].

Bacterial inoculation was carried out on two stages: (1) freeze-dried bacterial cultures were rehydrated and enriched in nutrient broth to a high starting concentration of approximately $10^8 \div 10^9$ cells/mL; (2) then, the inoculums were transferred and cultivated in fresh media for the achievement of working concentrations (details in Table 1). The new bacterial suspensions were directly treated or inoculated on agar plates firstly, left for 30 min, and then treated. After treatment, the samples on agar plates were incubated for 24 h at 37 °C; the liquid samples were diluted ten-fold and then spread on nutrient agar and cultivated at same conditions. The initial cell density and cell number in liquids were estimated by direct microscopic counting in the Bürker chamber (results presented as cells/mL) or by the count plate technique (results presented as colony forming units/mL). The data are means of three or more replicates. For better presentation of the plasma effect on the cells, pictures on scanning electron microscope were acquired.

**Table 1.** Experimental design for assessment of the bactericidal effect of the cold atmospheric argon plasma torch.

| Treated Media | Used Microorganisms | Treatment Conditions | Indicators for Assessment | |
|---|---|---|---|---|
| Surfaces: bacterial thick films on nutrient agar plates | *Pseudomonas aureofaciens* AP-9 (now: *chlororaphis*)-aerobic, Gram-negative, motile, polar-flagellated, rod-shaped bacterium | Time: 5, 10, 15, 20 s Wave power: 17 and 20 W Bacterial densities: $90 \times 10^5$ cells/cm$^2$ and $3 \times 10^5$ cells/cm$^2$ | (1) | Appearance of sterilization zones (zones without bacterial growth) |
| | | | (2) | Diameter of sterilization zones |
| Liquids: bacterial suspensions in nutrient broth media (exponential phase of bacterial growth) | (1) *Pseudomonas aureofaciens* AP-9 (now: *chlororaphis*) (2) *Brevibacillus laterosporus* BT-271-rod-shaped, Gram-positive, endospore-forming bacterium | Time: 10, 30, 60 s Wave power: 20 W Bacterial densities: $52 \times 10^7$ cells/mL (for *Pseudomonas*) and $76 \times 10^7$ cells/mL (for *Brevibacillus*) Volume of treated suspensions: 35 mL | (1) | Survival curves (number of colony forming units (CFUs) per volume versus treatment time) |
| | | | (2) | Bactericidal effect (log of the number of living bacteria in control versus the number of living cells after treatment) |
| | | | (3) | % bacteria killed |

### 2.2.3. Plasma Treatment of Bacterial Films on Surfaces

The results from surface sterilization experiments are presented in pictures of treated agar plates for visual estimation of the plasma effect (Figure 1).

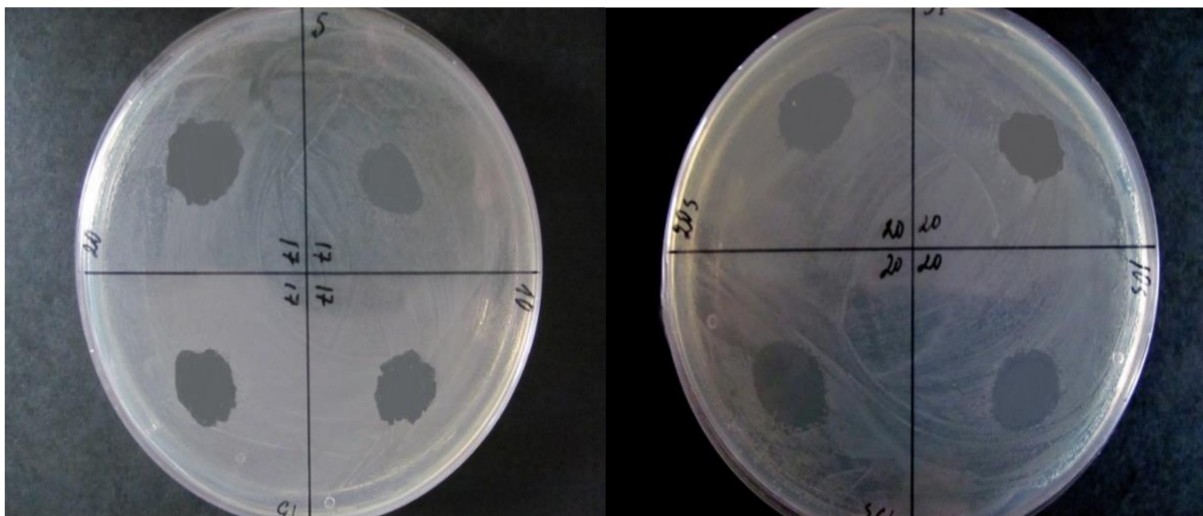

**Figure 1.** Photographs of sterilization zones on agar plates with $3 \times 10^5$ cells/cm$^2$ bacterial density of *Pseudomonas aureofaciens AP-9* at 17 W (**left**) and 20 W (**right**) wave power. The different sectors on Petri plates were treated with different durations—top-left 5 s, bottom-left 10 s, bottom-right 15 s, top-right 20 s.

The pale cover over all the surface of plates was bacterial film. The dark zones in the centers of each sector were treated with the plasma torch and they were without bacterial growth. The used plasma source had a clear sterilization effect on films of *Pseudomonas* AP-9, even at the shortest treatment time of 5 s. The boundaries of the zones were differentiated and the sterilization effect at this bacterial number ($3 \times 10^5$ cells/cm$^2$) was very stable. Our observations showed the absence of bacterial growth for more than 168 h in the zones, which maintained their sizes and purity.

The diameter and area of the sterilization zones depend on different factors; our results identify the following as the most important: treatment time, wave power and density of

treated bacterial films. Figure 2 presents the dependence of the sterilization zone area on these factors. The area is larger when the exposure time and wave power are increased, but high bacterial density has a limiting action for the achievement of effective and stable sterilization. At the cell density of $90 \times 10^5$ cells/cm$^2$, the diameter of obtained clear zones was smaller. The cell density of treated films had one of the leading roles in the achievement of effective decontamination. However, in all cases, the zone diameter was significantly larger than the diameter of the plasma torch. The generated plasma torch was 1–2 mm; the sterilization zones had a diameter of more than 5 mm. The temperature of plasma was close to room temperature and the effect of high temperature for bacteria-killing could be excluded. Probably, the diffusion of reactive species and the UV irradiation produced in the plasma torch made the sterilization zones much larger than the plasma contact area.

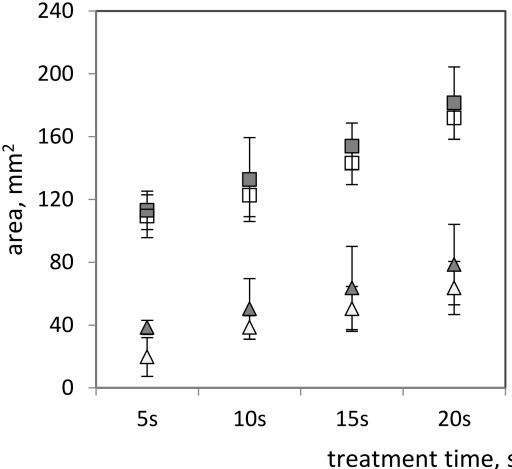

**Figure 2.** The area of sterilization zones as a function of treatment time at different wave powers (white markers—17 W; dark markers—20 W) and bacterial densities (triangle markers—$90 \times 10^5$ cells/cm$^2$; square markers—$3 \times 10^5$ cells/cm$^2$).

The data show that increasing the microbial density leads to a minimization of the plasma efficiency. One of the possible explanations is a formation of multilayered cellular structures at high bacterial densities—the upper layers have a protective action and the increased biomass limits the penetration of the plasma's active components in depth [56].

2.2.4. Plasma Decontamination of Liquids

It is proven that discharges in liquid media and on air-liquid borders generate active components in different concentrations and proportions compared to plasma surface treatment [72]. The intensive UV radiation and reactive oxygen and nitrogen species are the active agents, with the highest contribution to total bactericidal effect, while the actions of other plasma components are shielded from the liquid. The plasma efficiency varies depending on the mode of bacterial life as well, such as the biofilm or planktonic state of the treated bacteria [73,74].

In Figure 3, we present the bacterial inactivation in liquids of two bacterial strains: Gram-negative *Pseudomonas aureofaciens* AP-9 and Gram-positive *Brevibacillus laterosporus* BT-271. At the treated bacterial density of $10^7$ cells/mL and exposure times < 1 min, we did not achieve the complete sterilization of treated suspensions; only partial disinfection was registered. The inactivation was 99% or 2-Log reduction. The observed survival curves for both bacterial strains (Figure 3 left) have a similar biphasic shape with "tailing". This shape is typical for spore survival curves [75] and there are several concepts for its explanation. Cerf's model supposes the existence of subpopulations with different resistance to disinfectant; the protection role of dead cells and the products of their destruction for other more resistant cells also have an important role, maybe in combination with inactivation

of bactericidal effect at depth [76]. The treatment of 10 s leads to a quick reduction in the number of *Pseudomonas*, but increased exposure time has no significant bactericidal effect, and the survival curve has a level tailing (a zero slope of tailing). The *Brevibacillus* curve has a slope tailing and the reduction of the bacterial number is smoother. The values for the bactericidal effect (Figure 3 middle) and percentage of bacteria killed (Figure 3 right) indicate that the short plasma treatment more effectively inactivates the *Pseudomonas* cells than *Brevibacillus* cells. The bactericidal effect for *Pseudomonas* stays around 2.3 for all studied treatment durations, but for *Brevibacillus* it increases with treatment time from 1.1 to 2.6. The achieved percentage of bacteria killed for Pseudomonas is more than 99% at 10 s; for *Brevibacillus*, this value was attained with at least 30 s.

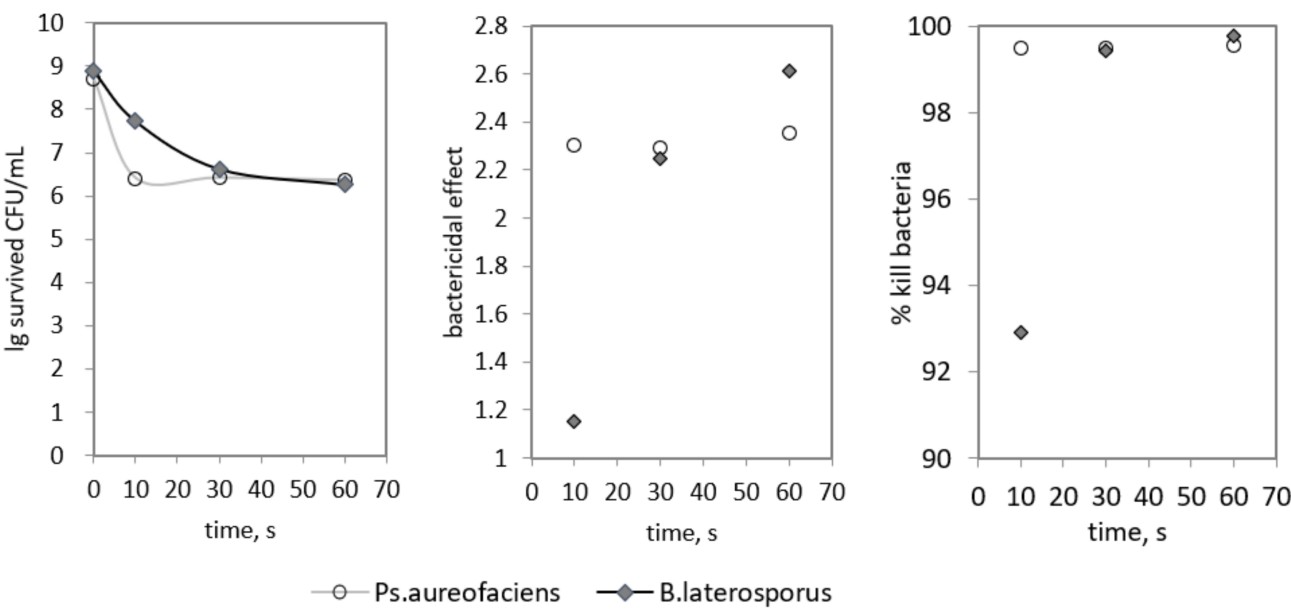

**Figure 3.** Survival curves of *Pseudomonas aureofaciens* AP-9 and *Brevibacillus laterosporus* BT-271 (**left**), the bactericidal effect of plasma (**middle**), and percentage bacteria killed (**right**) at different treatment times.

To prove the destructive effect of cold argon plasma on bacterial cells, we use a scanning electron microscope. The cell morphology of *Brevibacillus laterosporus* BT 271 (control and plasma-treated) is presented in the SEM micrographs (Figure 4). After plasma treatment with 30 s duration, no morphological change compared to the control was observed visually on the pictures, but the survival curve showed a significant decrease in the number of viable cells (Figure 3). Possibly, the initial interaction of plasma with prokaryotic Gram-positive cells is not characterized by external damages to the cell surface. With the extension of the exposure time to 60 s, a fraction of cell debris, small fragments, and leakage of cellular material were observed. The bacterial cells were deformed, with destroyed cell walls, serious damage on the surface, and loss of structural integrity. In Figure 5, the results of SEM micrographs of untreated and treated *Pseudomonas aureofaciens* AP-9 are presented. At 30 s treatment, the cells have severe injuries and the micrographs prove the greater sensitivity of Gram-negative bacteria to plasma exposure. The loss of cell viability in both bacterial types is probably due to a different leading mechanism of plasma-cell interaction. Laroussi et al. supposed that the major mechanism in Gram-positive bacteria is the diffusion of reactive species through the cell membranes and direct reaction with biomolecules [39].

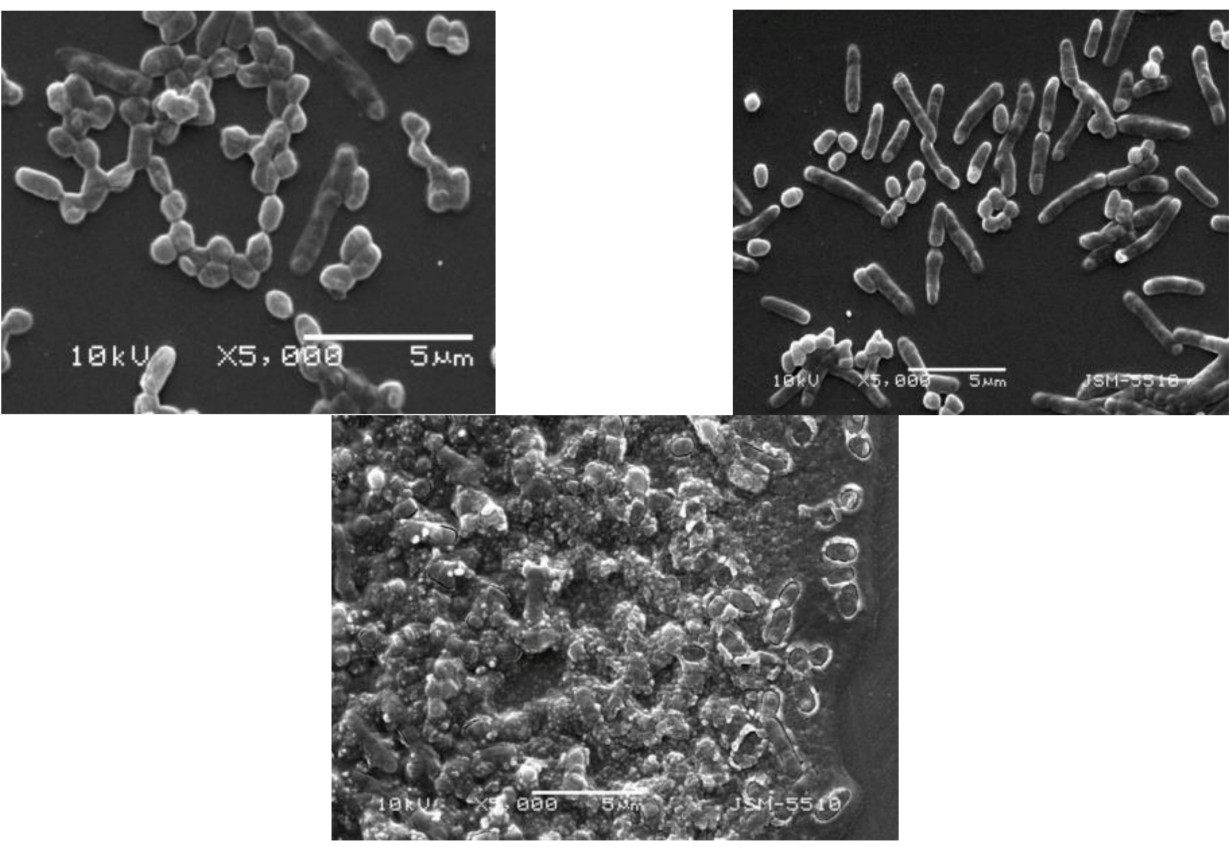

**Figure 4.** SEM micrographs of *Brevibacillus laterosporus* BT 271 (×5000, scanning electron microscope JEOL JSM 5510): (**top left**)—control; (**top right**)—plasma-treated sample with a duration of 30 s; (**below**)—plasma-treated sample with a duration of 60 s.

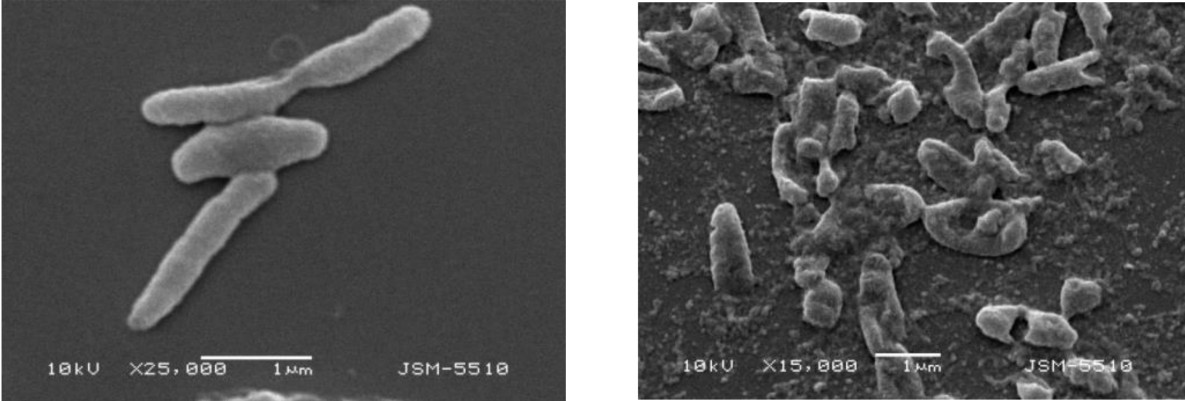

**Figure 5.** SEM micrographs of *Pseudomonas aureofaciens* AP-9 (×25,000 and 15,000, scanning electron microscope JEOL JSM 5510): (**left**)—control; (**right**)—plasma-treated sample with a duration of 30 s.

Our results are in line with the data of other authors for physical destruction of bacterial cells and spores after plasma treatment, and more visible damages on Gram-negative bacteria [39,59,77–80], but few studies have achieved a similar loss of cell integrity in Gram-positive bacteria with such a short exposure time.

### 3. Low-Temperature, Non-Equilibrium Plasma for Removal of Hazardous Chemicals

*3.1. A Brief State of the Art*

The growing production, wide use, and subsequent discharge of different toxic and persistent pollutants in the environment are another serious threat to human and ecosystem health. The effective solution of associated environmental problems needs adequate pollution control based on innovative approaches, and plasma technologies are one of the promising new alternatives. Considerable attention of researchers has been focused on plasma-assisted removal of volatile organic compounds from polluted air, and significant results have already been achieved in this area [81–84]. In the last years, these technologies have been applied for the treatment of gases from animal production to food processing facilities, and a high removal and energy efficiency has been achieved [85]. Solid waste and soil treatment is another area of research interest where the focus is mainly on thermal plasma application, but some types of non-thermal plasmas have been used [86,87]. Our efforts in the environmental field of plasma application are related to another target component—water—and its treatment for the removal of organic pollutants. The strong oxidation features of some highly energetic plasma components mean that plasma can be used successfully for the removal of unacceptable organic compounds in wastewater [88–92]. In many advanced strategies for pollution control and management, plasma technologies are considered as one of the options for the substitution of expensive traditional chemical methods in water treatment and the elimination of secondary pollution in treated water. High efficiency has already been achieved in the removal of phenol and a wide range of phenolic derivates [89,92–95], azo-dyes [86], pharmaceutical and antibiotic compounds [96–98], and carbon from plastic waste [99]. Several configurations have been developed and applied for the removal of different organic pollutants during discharge in the gas, liquid, or hybrid liquid-gas phase, such as pulsed high-voltage electrical discharge, glow discharge, and gliding arc [92].

*3.2. Case Study—Tretment of Organic Pollutants*

3.2.1. Characterization of the Plasma Generation System

The plasma source and characteristics of generated plasma were the same as those described above in Chapter 2.2.1. Discharge conditions are plasma radius 0.05 cm ÷ 0.1 cm; gas flow rate 3.0 ÷ 3.5 L/min. Input wave power is 25 ÷ 30 W.

3.2.2. Experimental Design

In the experimental design, we included treatment of waters with two bio-recalcitrant organic pollutants: phenol with an initial concentration of 500 mg/L (corresponding to Chemical Oxygen Demand COD = 1900 mgO$_2$/L), and azo-dye amaranth with a concentration of approximately 25 mg/L. The volume of treated samples was the same at 35 mL; the time of treatment varied from 30 s to 20 min. The concentration of pollutants was measured before and after treatment, according to standard spectrometric methods for the determination of COD by potassium dichromate and by the reduction of amaranth absorbance at 520 nm.

3.2.3. Reduction of Pollutant Concentration after Plasma Treatment

Figure 6 presents the reduction in the concentration of phenol and amaranth after plasma treatment. The concentration of these two organic compounds was reduced depending on treatment time. The 2 min treatment led to 45% removal of phenol in model wastewater, assessed by reduction of organic content of water; an increase of the time to 10 min achieved 62% removal. In the other model wastewater, the concentration of the xenobiotic compound amaranth decreased gradually with the increase of exposure time. The percentage of removal was 72% with 20 min treatment.

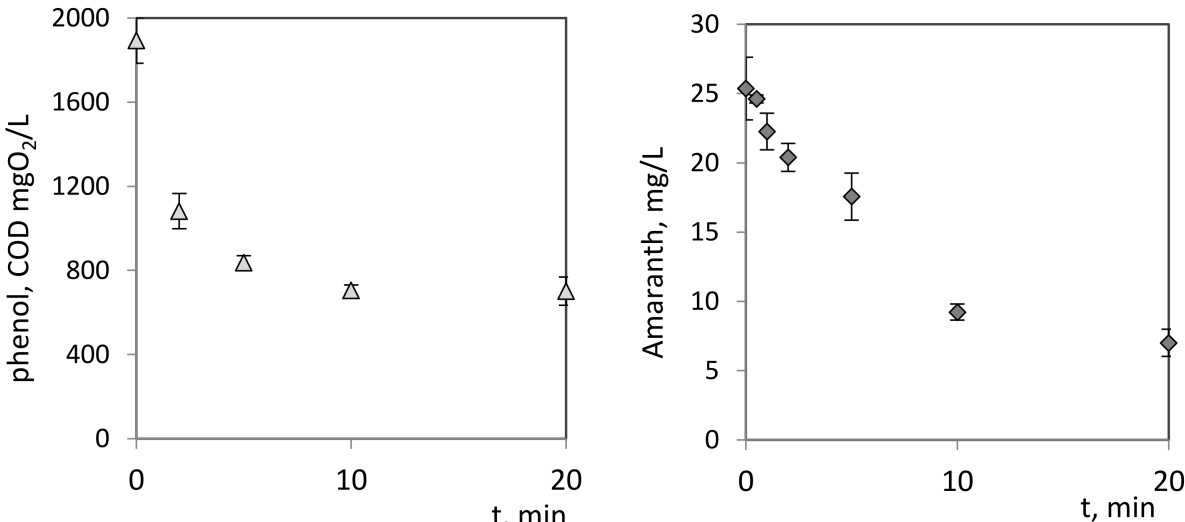

**Figure 6.** Variation in concentration of xenobiotic organic compounds—phenol, measured as COD (**left**), and amaranth (**right**)—after cold plasma treatment.

These results represent a preliminary assessment of the potential of microwave plasma to be used for the removal of organic residues and pollutants. Although the times used failed to achieve complete removal of pollutants, the data show that, even with such a short impact, there was sufficient effect to continue research in the environmental field with this type of plasma source.

## 4. Summary of Cold Atmospheric Plasma Potential for Microbial Decontamination and Chemical Removal with Future Perspectives in the Circular Economy Context

The need for a highly efficient, low-cost but reliable, biocompatible, and environmentally -friendly method for the removal of undesirable compounds now is very urgent in the growing implementation of the circular economy model. These targeted actions are required for the mitigation of risks related to the presence of hazardous microorganisms or substances in waste streams, and their potential entrance into new processes. Non-thermal atmospheric plasma has recently attracted great research interest as an innovative alternative for the operative solution of some acute problems related to safety and quality control, such as inactivation of hazardous microorganisms and viruses, and the degradation of toxic chemical substances.

The overview of some important contributions in the field proves the great potential of cold atmospheric plasma for bacterial inactivation in a variety of configurations (see the previous sections for references). Our results show the effective decontamination of surfaces by microwave plasma and the large bactericidal effect against *Pseudomonas* biofilm at very short exposure times. The data for the achievement of satisfactory and reliable sterilization against bacterial biofilms have been reported by many authors [53,54,56,78,80], but these studies, as well as ours, are primarily focused on monocultural or bicultural biofilms (biofilm from one or two different microbial cultures). Multispecies biofilms, which are the dominant type in the environment and cause many of the problems with unacceptable bacterial growth in different industrial devices, deserve special interest in upcoming plasma researches. The wide range of susceptible materials for bacterial biofilm formation must also be included in these future studies.

Different authors have compared the bactericidal effect of CAP in vitro and in vivo, and the results obtained showed a distinct mode of interaction. Dijksteel et al. have studied the safety and efficiency of CAP against *Pseudomonas aeruginosa*, strain PAO1. The used device—a flexible surface Dielectric Barrier Discharge—had an excellent bactericidal effect in vitro, but the in vivo experiments (rat wound infection model) demonstrated a limited bactericidal efficacy [100]. The authors suggested that environmental factors such as biofilm

formation could have played a negative role in the limited efficacy of CAP in vivo. Cooley et al. have proven the high bactericidal effectiveness of indirect non-thermal atmospheric plasma discharge on the biofilms of the same bacterial strain (PAO1), but the authors reported less effectiveness in the penetration of plasma below the targeted surface [101].

Treatment of bacterially-contaminated liquids is another very important area in plasma studies. Successful disinfection was attained with very high bacterial densities and a small timescale. We present SEM proofs for one of the possible mechanisms of plasma interaction with living prokaryotic cells: the physical destruction of bacterial cells and spores and the leaching of cell content after plasma treatment. Our results confirmed the more visible damages on cell surfaces of Gram-negative bacteria compared to Gram-positive, but we have also achieved a significant loss of cell integrity in Gram-positive bacteria with a short exposure time of <1 min.

The oxidizing components in non-thermal plasma determine its wide variety of environmental applications, as pre- or post-treatment steps for removal of recalcitrant organics in water, the treatment of polluted air, soil remediation, or the treatment of solid wastes in different waste streams. We studied the plasma-induced removal of phenol and azo dye amaranth from liquids. The tested treatment times in our experiments are not sufficient for 100% removal of organic compounds, but give a good base for further work and prove the potential of the used plasma source for broad environmental application, especially for the removal of chemical residues in different materials.

The most important future challenge in the field of plasma use in the circular economy context is related to the urgent need for the commercialization of these technologies. The efforts of all research groups working in the field of cold atmospheric plasma application (a small part of whose results we have been able to summarize here) have led to the elucidation of different aspects of the plasma scientific base. However, there is still a significant gap in plasma implementation for real practice, especially in industry and agriculture. This process is additionally hampered by the need for optimization of plasma-generating devices for each specific application. Nevertheless, cold atmospheric plasma has a high proven capacity for solving the urgent problems of microbial and chemical safety, and its successful wide implementation in practice is a matter of time.

**Author Contributions:** The concept of the paper was proposed during a personal discussion between all authors. The experimental design was proposed by Y.T. (Yovana Todorova), P.M., E.B., Y.T. (Yana Topalova); experiments were performed by Y.T. (Yovana Todorova), I.Y., P.M., T.B. and E.B. The original draft preparation was done by Y.T. (Yovana Todorova), E.B., T.B. and P.M. Writing—review and editing, Y.T. (Yovana Todorova), E.B. and Y.T. (Yana Topalova); supervision, Y.T. (Yana Topalova); funding acquisition, Y.T. (Yana Topalova). All authors have read and agreed to the published version of the manuscript.

**Funding:** This research was funded by Grant No. BG05M2OP001-1.002-0019: "Clean Technologies for Sustainable Environment-Waters, Waste, Energy for a Circular Economy", financed by the Science and Education for Smart Growth Operational Program (2014-2020) and co-financed by the EU through the European structural and investment fund.

**Informed Consent Statement:** Not applicable.

**Data Availability Statement:** The data that support the presented results of this study are available from the corresponding author upon reasonable request.

**Acknowledgments:** The authors are thankful to Neli Zheleva and Nikolay Dimitrov for their valuable help with SEM micrographs.

**Conflicts of Interest:** The authors declare no conflict of interest. The funders had no role in the design of the study; in the collection, analyses, or interpretation of data; in the writing of the manuscript, or in the decision to publish the results.

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
