# Peer review of "Non-Thermal Atmospheric Plasma for Microbial Decontamination and Removal of Hazardous Chemicals: An Overview in the Circular Economy Context with Data for Test Applications of Microwave Plasma Torch"

_processes, doi:10.3390/pr10030554_

Round 1

Reviewer 1 Report

The review paper is well written and the topic is well chosen, albeit the size of the manuscript is a bit synthetic.  I suggest to include a deeper and broader comparison with earlier literature, such as:

Safety and bactericidal efficacy of cold atmospheric plasma generated by a flexible surface Dielectric Barrier Discharge device against Pseudomonas aeruginosa in vitro and in vivo. Dijksteel GS, Ulrich MMW, Vlig M, Sobota A, Middelkoop E, Boekema BKHL.Ann Clin Microbiol Antimicrob. 2020 Aug 19;19(1):37. doi: 10.1186/s12941-020-00381-z.

Shock. 2020 Nov;54(5):681-687.  doi: 10.1097/SHK.0000000000001583.

Indirect, Non-Thermal Atmospheric Plasma Promotes Bacterial Killing in vitro and Wound Disinfection in vivo Using Monogenic and Polygenic Models of Type 2 Diabetes (Without Adverse Metabolic Complications)

Chris R Cooley 1John Mark McLain 1Samuel D Dupuy 1Adrianna E Eder 1Molly Wintenberg 2Kimberly Kelly-Wintenberg 2Alan Wintenberg 2J Jason Collier 3Susan J Burke 3Michael D Karlstad 1 DOI: 10.1097/SHK.0000000000001583   Front. Microbiol., 02 April 2019 | https://doi.org/10.3389/fmicb.2019.00622A Review on Non-thermal Atmospheric Plasma for Food Preservation: Mode of Action, Determinants of Effectiveness, and Applications Mercedes López1,2*, Tamara Calvo1, Miguel Prieto1,2, Rodolfo Múgica-Vidal3, Ignacio Muro-Fraguas3, Fernando Alba-Elías3 and Avelino Alvarez-Ordóñez   Front. Phys., 01 June 2021 | https://doi.org/10.3389/fphy.2021.683118Non-Thermal Plasma as a Novel Strategy for Treating or Preventing Viral Infection and Associated Disease Hager Mohamed1,  Gaurav Nayak2, Nicole Rendine3, Brian Wigdahl1, Fred C. Krebs1, Peter J. Bruggeman2 and Vandana Miller

Author Response

We would like to thank the Reviewer for the efforts and for helpful comments that will improve the quality of our manuscript. We have done our best to respond to the comments and recommendations. As indicated below, we have checked all the comments provided by the Reviewer and have made necessary changes according to their indications (mark with Track Changes).

Comment:

"I suggest to include a deeper and broader comparison with earlier literature, such as:

  1. Safety and bactericidal efficacy of cold atmospheric plasma generated by a flexible surface Dielectric Barrier Discharge device against Pseudomonas aeruginosa in vitro and in vivo. Dijksteel GS, Ulrich MMW, Vlig M, Sobota A, Middelkoop E, Boekema BKHL.Ann Clin Microbiol Antimicrob. 2020 Aug 19;19(1):37. doi: 10.1186/s12941-020-00381-z.
  2. Indirect, Non-Thermal Atmospheric Plasma Promotes Bacterial Killing in vitro and Wound Disinfection in vivo Using Monogenic and Polygenic Models of Type 2 Diabetes (Without Adverse Metabolic Complications) Chris R Cooley 1, John Mark McLain 1, Samuel D Dupuy 1, Adrianna E Eder 1, Molly Wintenberg 2, Kimberly Kelly-Wintenberg 2, Alan Wintenberg 2, J Jason Collier 3, Susan J Burke 3, Michael D Karlstad 1 DOI: 10.1097/SHK.0000000000001583 SHOCK, Vol. 54, No. 5, pp. 681–687, 2020
  3. Review on Non-thermal Atmospheric Plasma for Food Preservation: Mode of Action, Determinants of Effectiveness, and Applications Mercedes López1,2*, Tamara Calvo1, Miguel Prieto1,2, Rodolfo Múgica-Vidal3, Ignacio Muro-Fraguas3, Fernando Alba-Elías3 and Avelino Alvarez-Ordóñez Microbiol., 02 April 2019 | https://doi.org/10.3389/fmicb.2019.00622A
  4. Non-Thermal Plasma as a Novel Strategy for Treating or Preventing Viral Infection and Associated Disease Hager Mohamed1†, Gaurav Nayak2†, Nicole Rendine3, Brian Wigdahl1, Fred C. Krebs1‡, Peter J. Bruggeman2‡ and Vandana Miller Front. Phys., 01 June 2021 https://doi.org/10.3389/fphy.2021.683118 "

Answer:

Thank you for your suggestions. We included all of the suggested references in the text and according to our opinion, the discussion was enriched and deepened.

  1. The first and second suggested papers were discussed in section 4. Summary of cold atmospheric plasma potential for microbial decontamination and chemical removal with future perspectives in the circular economy context and the references were cited in the reference list as numbers 100 and 101.
  2. The third and fourth papers were cited in section 2. Low-temperature non-equilibrium plasma for microbial decontamination and the references were cited in the reference list as numbers 68 and 51, respectively.

The added new text is marked in the manuscript with Track Changes for easy orientation.

Reviewer 2 Report

This paper gives a good overview of plasma technologies for circular economy. I have just some questions for Figure 3.

  1. The treated bacterial density is 107 cells/mL in the text but the initial density in Fig. 3-left is 109 CFU/mL. “cells/mL” has different meaning with “CFU/mL”?

  1. How large are the error bars if you do the inactivation experiments several times?

Author Response

We would like to thank the Reviewer for the efforts and for helpful comments that will improve the quality of our manuscript. We have done our best to respond to the comments and recommendations. As indicated below, we have checked all the comments provided by the Reviewer and have made necessary changes according to their indications (mark with Track Changes).

Comment:

"This paper gives a good overview of plasma technologies for circular economy. I have just some questions for Figure 3.

  1. The treated bacterial density is 107 cells/mL in the text but the initial density in Fig. 3-left is 109 CFU/mL. “cells/mL” has different meaning with “CFU/mL”?
  2. How large are the error bars if you do the inactivation experiments several times?"

 Answer:

  1. The bacterial density of the suspensions used in the experiments to evaluate the plasma effect on surfaces for the initial inoculation was estimated by direct cell counting. This is the direct microscopic method and the results are in cells/mL. After inoculation of agar plates with bacteria and their treatment with plasma, the effect was determined by the plate count. Unlike in direct microscopic counting where all cells, dead and living, are counted, CFU measures viable cells. The unit CFU estimates the number of bacteria in a sample that are viable, able to multiply via binary fission under controlled conditions. Counting with colony-forming units requires culturing the microbes and counts only viable cells, in contrast with the microscopic examination which counts all cells, living or dead. Since we have used both methods, we consider it more correct to use both units of measurement. For more clarity, we have added a few explanations in the text on page 7, immediately after Table 1 (marked with Track changes).
  2. In Figure 3, we presented the survival curves of studied bacteria, the bactericidal effect of plasma and % killed bacteria at different treatment times. Survival curves in microbiology are usually presented on a logarithmic scale, the other parameters are calculated from the first. We think it is not correct to put the error bars in these cases. But we would like to thank the Reviewer for pointing us to the fact that we forgot to put the error bars in Figure 2. We replace the figure with identical but with error bars. Thank you again.

Reviewer 3 Report

The manuscript is focused on providing a comprehensive review of multiple applications of low-temperature atmospheric plasma discharges for disinfection and decontamination of surfaces and liquids. After careful evaluation, I found the manuscript to be well-written: it is logically structured, the experiments are meticulously planned and executed, the results are interpreted well and supported by appropriate references or arguments. The evaluation of experimental findings is presented and the line of arguments is easy to follow. All the conclusions are supported adequately by the data presented. I do not have scientific criticism and, thus, I recommend this work for publication in the Processes journal.

Author Response

Comment:

"The manuscript is focused on providing a comprehensive review of multiple applications of low-temperature atmospheric plasma discharges for disinfection and decontamination of surfaces and liquids. After careful evaluation, I found the manuscript to be well-written: it is logically structured, the experiments are meticulously planned and executed, the results are interpreted well and supported by appropriate references or arguments. The evaluation of experimental findings is presented and the line of arguments is easy to follow. All the conclusions are supported adequately by the data presented. I do not have scientific criticism and, thus, I recommend this work for publication in the Processes journal."

We would like to thank the reviewer for evaluating our work and efforts. We hope that we have improved the manuscript quality even more.